# Comparative analysis of various clinical specimens in detection of SARS-CoV-2 using rRT-PCR in new and follow up cases of COVID-19 infection: Quest for the best choice

Kuldeep Sharma[1‡], Pragya Aggarwala[1‡], Deepa Gandhi[1‡], Anuniti Mathias[1‡], Priyanka Singh[1], Somya Sharma[1], Sanjay Singh Negi [1]*, Anudita Bhargava[1], Padma Das[1], Ujjwala Gaikwad[1], Archana Wankhede[1], Ajoy Behra[2], Nitin M. Nagarkar[3]

**1** Microbiology Department, AIIMS, Raipur, Chhattisgarh, India, **2** Department of Pulmonary Medicine, AIIMS, Raipur, Chhattisgarh, India, **3** AIIMS, Raipur, Chhattisgarh, India

‡ These authors contributed equally to this work as co-first authors.
* negidr@yahoo.co.in

**Data Availability Statement:** The study contains the potential information of identity of the patients and thus cannot be shared on direct request to

## Abstract

### Background

An appropriate specimen is of paramount importance in Real Time reverse transcription-polymerase chain reaction (rRT-PCR) based diagnosis of novel coronavirus (nCoV) disease (COVID-19). Thus, it's pertinent to evaluate various diversified clinical specimens' diagnostic utility in both diagnosis and follow-up of COVID-19.

### Methods

A total of 924 initial specimens from 130 COVID-19 symptomatic cases before initiation of treatment and 665 follow up specimens from 15 randomly selected cases comprising of equal number of nasopharyngeal swab (NPS), oropharyngeal swab (OPS), combined NPS and OPS (Combined swab), sputum, plasma, serum and urine were evaluated by rRT-PCR.

### Results

Demographic analysis showed males (86) twice more affected by COVID-19 than females (44) (p = 0.00001). Combined swabs showed a positivity rate of 100% followed by NPS (91.5%), OPS (72.3%), sputum (63%), while nCoV was found undetected in urine, plasma and serum specimens. The lowest cycle threshold (Ct) values of targeted genes *E*, *ORF1b* and *RdRP* are 10.56, 10.14 and 12.26 respectively and their lowest average Ct values were found in combined swab which indicates high viral load in combined swab among all other specimen types. Analysis of 665 follow-up multi-varied specimens also showed combined swab as the last specimen among all specimen types to become negative, after an average 6.6 (range 4–10) days post-treatment, having lowest (15.48) and average (29.96) Ct values of *ORF1b* respectively indicating posterior nasopharyngeal tract as primary nCoV afflicted site with high viral load.

corresponding author. Any request for the data other than shown in the paper requires an official institutional request to IEC, AIIMS, Raipur to the address mentioned below: The Secretary Institutional Ethical Committee(IEC), Room No. 2103, 2nd floor, Medical College Complex, Gate No. 5, All India Institute of Medical Sciences (AIIMS), Raipur Chhattisgarh, India-492099. Phone NO.: +91 771-2577231 Fax No.: +91 771-2572999 Mail id: iec@aiimsraipur.edu.in.

**Funding:** This study was supported by AIIMS, Raipur, Chhattisgarh (AIIMS/Raipur/Admin/110).

**Competing interests:** The authors have declared that no competing interests exist.

## Conclusion

The combined swab may be recommended as a more appropriate specimen for both diagnosis and monitoring of COVID-19 treatment by rRT-PCR for assessing virus clearance to help physicians in taking evidence-based decision before discharging patients. Implementing combined swabs globally will definitely help in management and control of the pandemic, as it is the need of the hour.

## 1. Introduction

SARS-CoV-2, a name given to the nCoV by the International Committee of Taxonomy of Viruses (ICTV), was first reported in December 2019 from Wuhan, China. Since then, it has posed a devastating looming threat to the world, as around 216 countries and territories are so far affected by the virus causing the infection named COVID-19 [1]. Till the date of reporting on 11.11.2020, 50,810,763 were infected, and 1,263,844 succumbed worldwide to the infection [2]. India is the second most affected country after USA, with 8,636,011 confirmed cases and 127,571 deaths as of 11.11.2020 [2]. The disease can occur in any age-group, being more complicated and life-threatening in patients of the older age group and those with underlying co-morbid conditions such as diabetes, hypertension, cardiovascular and cerebrovascular disease [3–5]. The early clinical presentation of the COVID-19 varies from entirely asymptomatic to severely symptomatic. In India, more than 70% of the laboratory-confirmed cases are asymptomatic [3]. In symptomatic patients, the clinical manifestation includes fever, myalgia, dry cough, fatigue, productive cough, shortness of breath, chest pain, loss of taste and smell, etc. The radiological finding of ground-glass opacities on chest X-ray is one of the prominent observations [4, 5]. Since SARS-CoV-2 has a high human-to-human transmissibility rate, early diagnosis, immediate isolation and early treatment of positive patients are key to successful management of the pandemic by preventing its spread to others. Since testing is the corner stone of managing the COVID-19 pandemic, highly sensitive and specific testing is essentially required for early identification of not only the symptomatic cases but also of the asymptomatic cases and their close high-risk contacts, which would potentially break the chain of transmission of COVID-19 infection, which otherwise appears unstoppable at the moment.

Among various viral diagnostic modalities, virus isolation does not appear practically feasible for COVID-19 since it requires Biosafety Level (BSL)-3 laboratory, high technical expertise and longer turn-around time of 3–5 days to identify cytopathic effect in specific cell lines such as Vero E6 cells [6]. Serological tests based on SARS-CoV-2 antibody detection, have been reported with varying sensitivity (34 to 80%), cross reactivity with other SARS-CoV, varying rates of seroconversion between 7 and 11 days after onset of symptoms and varying immunological responses [7, 8]. Antigen detection assays also have the limitation of poor sensitivity and negative predictive values [7].

Therefore, rRT-PCR is the recommended reference test to establish laboratory diagnosis of SARS-CoV-2 by detecting at least two genes from various conserved region of specific structural Spike (*S*), Envelope (*E*), Nucleocapsid (*N*) genes and the nonstructural RNA dependent RNA polymerase (*RdRp*) and replicase open reading frame (*ORF*) 1a /b, *ORF 1b*-nsp14 [5, 7, 9]. Various in-house and commercially available rRT-PCR test kits are presently being used for identification of SARS-CoV-2 in the clinical specimens. OPS and/or NPS are currently the most preferred clinical specimens due to non-invasive and easily accessible nature and is being utilized across the globe to diagnose COVID-19 infection. During initial period of the pandemic

in Wuhan, NPS was used to detect SARS-CoV-2 [5]. Since then, various studies, systemic reviews and meta-analysis have evaluated the spectrum of clinical specimens in the quest of optimal specimen for its inclusion in guidelines for early identification of SARS-CoV-2 to provide timely treatment to prevent its transmission and thus better management of the pandemic [5, 10–18]. These include upper respiratory tract specimen (saliva, OPS, NPS, nasal swab), lower respiratory specimen {sputum, bronchoalveolar lavage (BAL), endotracheal aspirate (ET), fibrobronchoscope brush biopsy (FBB)}, blood and its products (serum, plasma), urine, feces and rectal swab. These studies and meta-analysis have various conclusions, probably because of analyzing a different spectrum of clinical specimens. Systemic review and meta-analysis by Bwire et al. [17] and study by Wang W et al. [14] reported the highest SARS-CoV-2 detection rate in BAL, while similar review and meta-analysis by Mohammadi et al. [18] and study by Zhang H et al. [13] recommends specimen of sputum for detection of SARS-CoV-2. Liu et al. [10] and Tong et al. [12] advocated NPS as specimen of choice for detection of nCoV. Rao et al. [11], on the other hand, found random saliva with a higher detection rate of nCoV than paired NPS and OPS swab. The optimal clinical specimens depend on various factors of ease of accessibility, non-invasive nature, a lesser risk to health care professional while collecting specimen and good viral loads for higher chances of detection. The collection of BAL, ET and FBB although have a higher detection rate and may be a specimen of choice in admitted pneumonia cases, yet it always poses a risk of generating aerosols to cause infection to healthcare workers. Additionally, they also cannot be a specimen of choice in managing pandemic infection of COVID-19 showing variable clinical manifestation from asymptomatic to mild/moderate and severe cases. Sputum, on the other hand, also pose a challenge not only for collection from cases of COVID-19 patients with dry cough but also for lower detection rate of nCoV as reported earlier [12]. Overall, there is certain uncertainty in understanding the specimens/sites from which the virus can be maximally diagnosed and which can be collected in field/community without posing health hazard to healthcare worker. Furthermore, these published studies have also not addressed optimal specimen in patients undergoing treatment to provide the appropriate prognostic indicator of viral clearance. Considering these facts, this study was undertaken to evaluate various clinical specimens that must be more accessible and feasible and can become a specimen of choice for early identification of SARS-CoV-2 for better management of COVID-19 pandemic. The proposed study has thus evaluated various specimens comprising of combined/ paired naso and oropharyngeal swab (hereafter referred to as a combined swab in the text), NPS, OPS, sputum, plasma, serum, urine and ET from known positive COVID-19 patients to understand their diagnostic utility in detection of SARS-CoV-2 as well as monitoring of follow-up cases of COVID-19 undergoing treatment. This study will also provide insight if this virus can also be transmitted in other ways than merely by respiratory droplets.

## 2. Methods

### 2.1. Patient selection

All India Institute of Medical Sciences (AIIMS)-Raipur is a designated tertiary-care hospital for diagnosis and treatment of COVID-19 patients in Chhattisgarh, a state in Central India. A total of 5000 suspected COVID-19 patients from May 2020 till June 2020, fulfilling either of the various testing criteria, laid down by the government of India, were referred to AIIMS, Raipur for diagnosis of COVID-19 infection by rRT-PCR test [19].

Among 5000-suspected patients, 137 outpatients were diagnosed for COVID-19 infection (2.7% positivity rate) by rRT-PCR using a combined swab. All these patients were subsequently admitted in the COVID ward of AIIMS, Raipur for isolation and treatment. These patients were evaluated in terms of the following inclusion and exclusion criteria.

**2.1.1. Inclusion criteria.** All suspected COVID-19 symptomatic patients were included in the study if fulfilling the following criteria-

1. Detected positive for COVID-19 infection by rRT-PCR.

2. Not on any anti-viral / anti-malarial (Hydroxychloroquine) / antibiotic (Azithromycin).

3. Admitted in COVID-19 ward of AIIMS, Raipur for treatment.

**2.1.2. Exclusion criteria.**

a. Nonfulfillment of any of the inclusion criteria.

Among them, 07 patients with a recent history of taking Azithromycin were excluded. Accordingly, only 130 patients were enrolled in the study after taking their consent. This study was approved by the Institutional ethical committee (IEC) of AIIMS, Raipur, Chhattisgarh (AIIMSRPR/IEC/2020/536).

Before starting a standard treatment regimen of Hydroxychloroquine and Azithromycin, all these patients were requested to provide clinical specimens of the following nature.

a. NPS

b. OPS

c. Combined (naso and oropharyngeal) swab

d. Sputum

e. Serum

f. Plasma

g. Urine

All swab specimens were collected from these patients before washing in morning by using sterile nylon flocked swab in viral transport medium (VTM) (HiMedia, India). An NPS was collected from a single nostril (posterior nasopharyngx) while OPS was collected from both sides of the throat. The combined swab of both NPS and OPS was collected in a single tube of VTM. In total, 910 (7 specimen types X 130 cases) specimens were tested by rRT-PCR. In addition, 14 ET were also obtained from an equal number of intubated patients. Thus, a total of 924 specimens were obtained from new patients prior to starting their treatment.

The positivity rate with all the seven types of clinical specimen was also tested in randomly selected 15 patients in their daily follow-up until the negative finding of rRT-PCR was achieved in two consecutive days' specimens of all seven types. Six hundred and sixty-five (665) follow-up specimens were collected from these 15 admitted patients. Thus, 924 initial and 665 follow-up specimens were tested by rRT-PCR for the identification of SARS-CoV-2.

## 2.2. RNA extraction

All the clinical specimens were processed for viral RNA isolation by using a commercially available QIAgen Viral RNA extraction kit, Germany, as per the manufacturer instructions. Briefly, 140μl of the specimen has been treated with 560μl of prepared buffer AVL containing carrier RNA (1 μg/μl). After brief pulse vortexing and 10-minutes incubation at room temperature, the specimen was precipitated by adding 560μl of pre-chilled ethanol. The treated specimen was then transferred to the spin column. Viral RNA was purified by consecutive treatment with 500μl of buffer AW1 and AW2. Finally, it was eluted in 60μl buffer AVE.

## 2.3. rRT-PCR test

This test was performed with primers and probes provided by Indian Council of Medical Research (ICMR), targeting *E*, *RdRP* and *ORF1b* genomic region of SARS-CoV-2 and internal control of human *RNAseP* as described earlier [20–22] (Table 1). Briefly, the 25 µl rRT-PCR reaction contained 12.5 µl 2x buffer, 1µl 25X RT-PCR enzyme mix (both from AgPath One-Step RT-PCR kit, ThermoFisher Scientific, USA), 1.5 µl Primer-Probe mix, 5 µl RNAse/DNase free sterile water and 5µl RNA template. The rRT-PCR test was carried out in CFX 96 Real Time PCR machine of Biorad, USA using the thermal cycling condition of 55°C for 30 min, 95°C for 3 min and 45 repeated cycles of 95°C for 15 sec and 58°C for 30 sec. The tested specimen was considered positive for SARS-CoV-2 for the cycle threshold (Ct) value less than or equal to 35 for *E* gene and both *RdRP* and *ORF1b* or either of *RdRP* or *ORF1b*. The positive and negative controls consisted of viral RNA plasmid and sterile nuclease-free water, respectively.

## 2.4. Gold standard

All 130 rRT-PCR detected cases of COVID-19 infection were considered as the known positive cases to evaluate the efficacy of various clinical specimens for diagnostic utility.

## 2.5. Statistical analysis

Categorical variables were analyzed by chi-square ($\chi^2$) and student t-test by using SPSS 16 version 18 (SPSS Inc., Chicago, IL, USA) to compare intergroup detection rate by considering $p<0.05$ statistically significant.

# 3. Results

A total of 130 known positive cases of COVID-19 infection were evaluated in their 924 clinical specimens obtained from different anatomical sites by rRT-PCR to detect SARS-CoV-2. Demographic analysis of these patients showed the median age of 40.14 years (range 5 to 74 years). Among them, 86 were males while 44 were females showing a significant higher COVID-19 infection rate in males than females ($\chi^2 = 27.13$, p = 0.00001, p<0.05). Median age calculated for males was 42.97 years, whereas, for females it was 32.07 years.

rRT-PCR detected all 130 cases with 100% positivity in combined swab (Table 2). NPS was the next appropriate clinical specimen showing a detection rate of 91.5%, followed by OPS and sputum specimens showing 72.3 and 63% positivity, respectively. None of the specimens of urine, plasma or serum showed detection of SARS-CoV-2. The 14 ET specimens showed 92.8% positivity by rRT-PCR. Combined swabs showed a significantly higher detection rate of SARS-CoV-2 in comparison to NPS, OPS and Sputum ($\chi^2 = 75.46$, p = <0.001, p<0.05). On comparison of various individual specimens with combined swabs, a significant difference was noticed in positivity rate between combined swab versus NPS ($\chi^2 = 11.48$, p = 0.0007, p<0.05), combined swab versus OPS ($\chi^2 = 12.68$, P = <0.001, p<0.05) and combined swab versus sputum ($\chi^2 = 58.86$ p = <0.001, p<0.05). NPS positive detection rate was also found to be significantly higher as compared to OPS and sputum specimen ($\chi^2 = 16.23$, p = 0.000056, p<0.05; $\chi^2 = 30.01$, p,0.00001, p<0.05). However, OPS positive detection rate was not found significantly higher than sputum positivity ($\chi^2 = 2.53$, p = 0.11, p>0.05).

Among individual swabs, NPS showed a higher detection rate than OPS. A total of 25 cases (19.2%, 16 males, 9 females) were detected in NPS but undetected in OPS (Table 2). However, a total of 11 (8.4%) cases were missed with NPS alone as the specimen, while nCoV was not detectable in 48 (36.9%) sputum specimens. No case was exclusively detected in OPS or sputum.

**Table 1. Primer sequence of various genes of SARS-CoV-2 for rRT-PCR.**

| Target gene | Sequence(5'-3') | Source |
|---|---|---|
| **E gene** | `ACAGGTACGTTAATAGTTAATAGCGT` | Corman et al. [20] |
| | `ATATTGCAGCAGTACGCACACA` | |
| | `FAM-ACACTAGCCATCCTTACTGCGCTTCG-BHQ` | |
| **RNaseP (Internal Control)** | `AGATTTGGACCTGCGAGCG` | CDC, 2020. [21] |
| | `GAGCGGCTGTCTCCACAAGT` | |
| | `FAM-TTCTGACCTGAAGGCTCTGCGCG-BHQ` | |
| **RdRp (Confirmatory)** | `GTGARATGGTCATGTGTGGCGG` | Corman et al. [20] |
| | `CARATGTTAAASACACTATTAGCATA` | |
| | `FAM-CAGGTGGAACCTCATCAGGAGATGC-QSY` | |
| **ORF1b (Confirmatory)** | `TGGGGYTTTACRGGTAACCT` | Poon et al. [22] |
| | `AACRCGCTTAACAAAGCACTC` | |
| | `FAM-TAGTTGTGATGCWATCATGACTAG-QSY` | |

The Ct (threshold cycle) values of *ORF1b*, *RdRP* and *E* gene were also compared between different clinical specimens (Fig 1). The cluttering of the Ct values was seen due to maximum Ct values falling between 20 and 32. The lowest Ct values of 10.56, 10.14 and 12.26 for *E*, *ORF1b* and *RdRP* respectively were obtained in combined swab followed by NPS, Sputum and OPS (Fig 1). The average Ct value of *E*, *ORF* and *RdRP* were 25.75, 26.94 and 27.06 in the combined swabs followed by NP, sputum and OP swabs respectively (Fig 2). The theoretical correlation of inverse relationship between Ct values and viral load imperatively indicates of higher viral load in specimen with low Ct and vise-versa. Thus, it can be inferred that maximum viral load was present in the combined swab, followed by NPS, sputum and OPS. The specimens of urine, serum and plasma did not show any sigmoidal amplification- based Ct values. The t-test comparison of average Ct value of all the targeted genes namely *E*, *ORF1b* and *RdRp* in various specimen categories showed a significant difference when the combined swab was compared individually with NPS ($p = 0.021$, $t = -2.315$), OPS ($p = 0.0003$, $t = -3.66$) and sputum ($p = 0.0027$, $t = -3.028$).

In randomly selected 15 follow up patients' testing, all seven types of specimens of combined swab, NPS, OPS, sputum, serum, plasma and urine were tested every day till the two consecutive days' rRT-PCR showed negative results in each specimen type (Figs 3 and 4 and Table 3). In the 'follow-up' category, a total of 665 specimens were obtained from 4 to 10 days after admission, with an average of 6.66 days (Fig 3). A gradual increase in Ct values of *ORF1b*

**Table 2. Positivity of rRT-PCR in different clinical samples in 130 known COVID-19 patients.**

| Total Patient (n = 130) | Combined swab (n = 130) No.(%)CI | NPS (n = 130) No.(%) CI | OPS (n = 130) No.(%) CI | Sputum (n = 130) No.(%) CI | Urine (n = 130) No.(%) CI | Plasma (n = 130) No.(%) CI | Serum (n = 130) No.(%) CI | Tracheal Aspirate (n = 14) No.(%) CI |
|---|---|---|---|---|---|---|---|---|
| **Male (n = 86)** | 86(100) (95.8–100) | 79(91.5) (83.9–96.6) | 63(72.3) (62.6–82.2) | 54(62.7) (51.7–72.9) | 0(0) | 0 | 0 | 13(92.8) (66.1–99.8) |
| **Female(n = 44)** | 44(100) (91.9–100) | 40(90.9) (78.3–97.4) | 31(70.4) (54.8–83.2) | 28(63.6) (47.8–77.6) | 0(0) | 0 | 0 | NA |
| **Total** | 130(100) (97.2–100) | 119(91.5) (85.3–95.7) | 94(72.3) (63.8–79.8) | 82(63.0) (54.2–71.4) | 0(0) | 0 | 0 | 13(92.8) (66.1–99.8) |

Tracheal aspirate was obtained from 14 male cases only. n (number tested), No. (Number), % (Percentage), CI (Confidence Interval), NA (No samples were obtained).

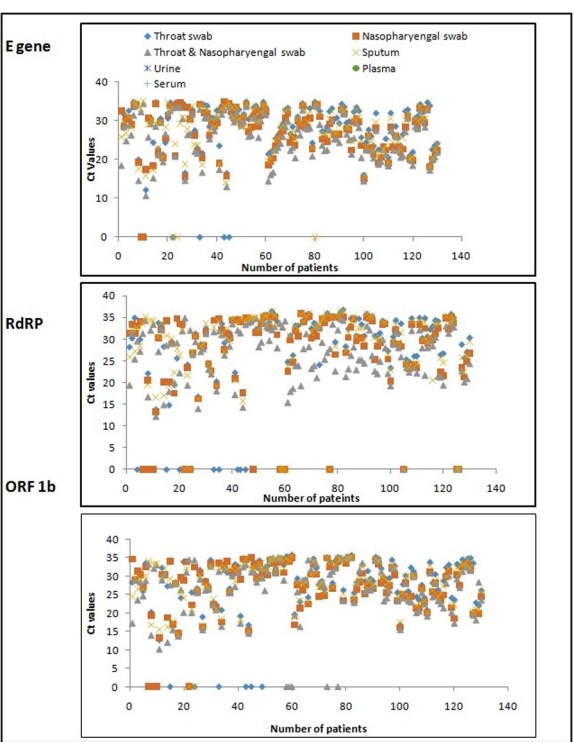

**Fig 1. The threshold cycle (Ct) values of *E*, *RdRP* and *ORF 1ab* region of SARS-CoV-2 in different clinical samples obtained from 130 patients.** The lowest Ct values of all the three target of *E*, *RdRP* and *ORF 1ab* were obtained in combined swab followed by NP, Sputum and Throat swab. Urine, Plasma and Serum samples have not shown any amplification.

from combined swab, NPS, OPS and sputum were noticed in daily testing indicating patients' affirmative response to treatment and virus clearance while other specimens of plasma, serum and urine showed no detection of SARS-CoV-2 (Fig 4). The maximum duration of days for clearance of virus was observed in combined swab (Fig 4 and Table 3). The earliest clearance with maximum detection of *ORF1b* was seen in patient P3 in which combined swab and NPS showed the presence of virus for only two treatment days and P11 in which only combined swab showed the presence of virus for two treatment days. Patients 1, 2, 7, 9, 11, 12, 13, 14 and 15 exclusively showed a longer duration of detection of nCOV in combined swab. Patient 10 shed virus in combined and NPS specimen for the longest period of nine days, followed by P7, which showed nCoV detection in only combined swab for consecutive seven days. During

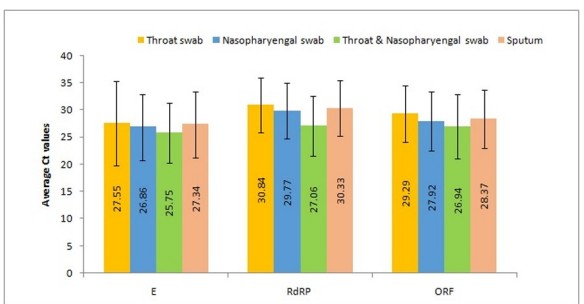

**Fig 2. The average Ct value of *E*, *RdRP* and *ORF 1ab* gene in different clinical samples.**

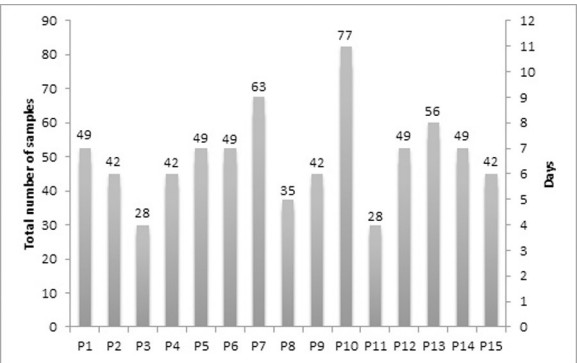

**Fig 3. Number of samples tested for 15 follow up cases till rRT-PCR showed negative results in two consecutive days sample.** Total number of samples per patients divided by 7 number of samples collected on daily basis gives the number of days the samples were collected for particular patients. Last two days 7 different types of samples were found negative for all the patients.

treatment monitoring, the average days of rRT-PCR positivity were 4.5, 3.7, 3.4 and 3.6 from the combined swab, NPS, OPS and sputum, respectively.

## 4. Discussion

The discharge policy for COVID-19 cases emphasized negative rRT-PCR findings on two consecutive days of respiratory specimen after symptom resolves. To give specific and accurate negative results, every laboratory needs to rule out false-negative PCR result, which otherwise would lead to discharge of such patient, leading to a high probability of transmission in the community, especially the family members and other close contacts. The importance of appropriate sampling in helping the laboratory to diagnose the COVID-19 infection accurately cannot be overemphasized. An appropriate specimen is the foundation stone for good laboratory test result and is one of the essential pre-analytical parameters for quality assurance. It is well-accepted fact that improper specimen is bound to generate an incorrect result. It is therefore said that '*garbage in will yield garbage out*'. The appropriate specimen must also be the optimal specimen in monitoring treatment/follow-up cases to help the clinician in management by taking evidence based decision on discharge. This study was thus conducted to analyze the most appropriate specimen for performing rRT-PCR to diagnose SARS-CoV-2 and monitor follow-up cases.

The present study showed differences in sensitivity of combined swab in comparison to NPS and OPS with which 8.2 and 19.2% positive cases were missed respectively. Thus, if tested alone, NPS and OPS may cause remarkable false-negative results that could lead to a discharge of these infected patients who are still shedding SARS-CoV-2 from their upper respiratory tract and may be a potential source for transmission of COVID-19 infection. We have compared various studies to assess their finding of clinical suitability of different biodiversified specimens (Table 4). In a study by Wang X et al. [23], it was observed that 73.1% of positive nasopharyngeal cases could not be detected with OPS. Our study exclusively noted that 19.2% of cases were detected by only combined swabs and were missed by other specimen types. The detection rate in sputum was significantly lower as compared to combined swab and individual NPS and OPS. Thus, sputum specimen alone for diagnosis of COVID-19 infection by rRT-PCR may not be recommended, as it missed 36.9% of cases in the present study. Our finding is also corroborated by earlier reported study showing a low positivity rate of 28.53% using sputum in detection of nCoV [12]. However, our finding of low positivity in sputum is in

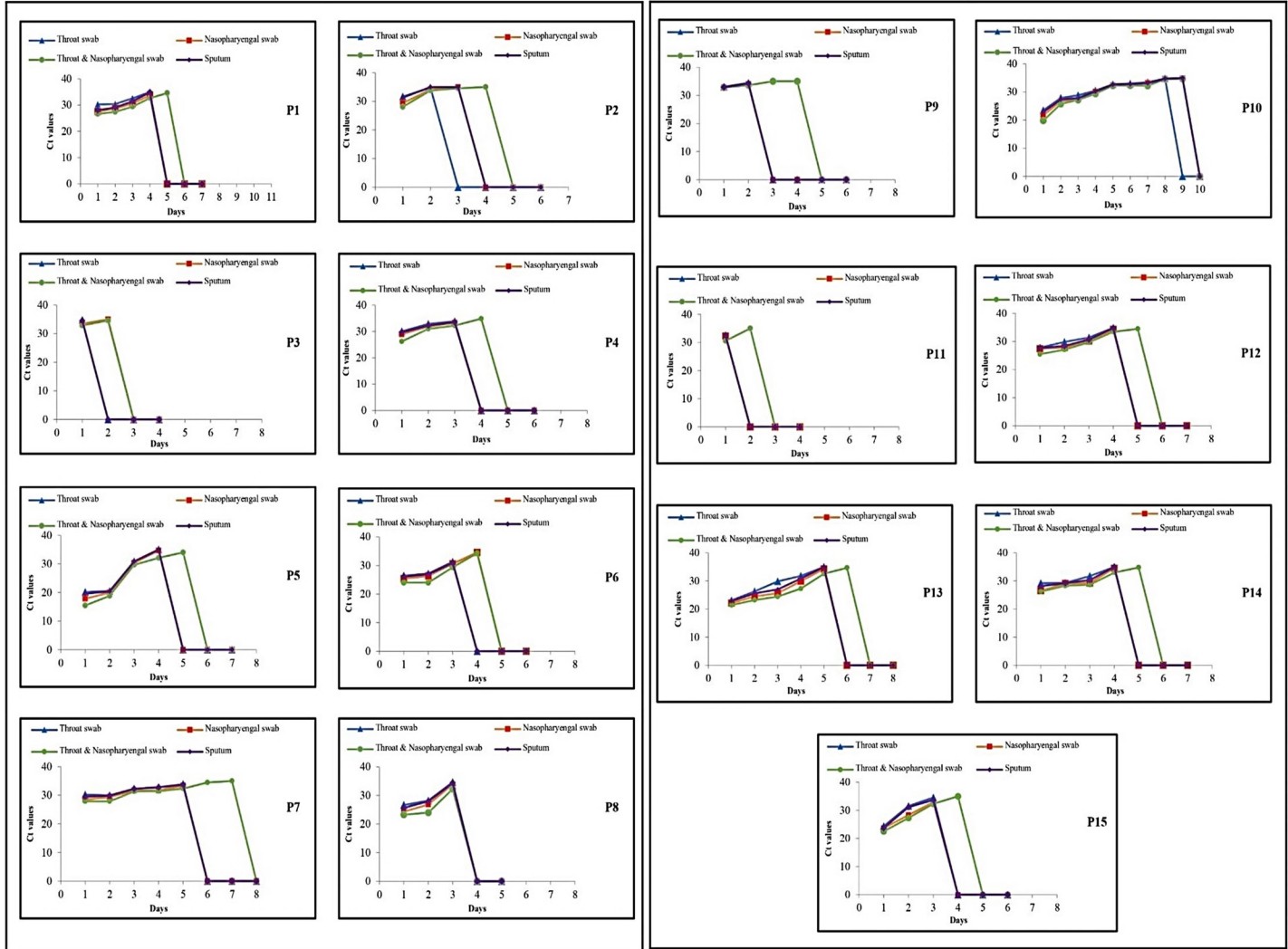

**Fig 4. Ct values of *ORF 1ab* in various clinical samples of 15 follow up cases.**

contrast to some of the earlier reported studies and meta-analysis. The systemic review and meta-analysis earlier had found sputum a better specimen than NPS and OPS [18, 19]. A separate study by Zhang H et al. [13] too found a higher detection rate of 79.2% in sputum than 37.5% and 20.8% positivity in NPS and OPS, respectively. Among sputum and OPS, Wang W et al. [14] found higher positivity with sputum, whereas Chan et al. [24] and Liu R et al. [25] did not find any significant difference in positivity between them. We further have the opinion of sputum being a non-ideal specimen in patients of COVID-19 infection with symptoms of dry cough and unable to produce sputum.

If only one swab is to be used for COVID-19 diagnosis, then NPS should be preferred over other specimens of OPS, sputum, serum, plasma and urine considering its higher detection rate of nCoV in our study. This preference is in line with the earlier finding of Tong et al. [12], who found a higher detection rate of nCoV in NPS than BAL, OPS, sputum, urine, blood, stool, anal swabs and corneal secretions. The findings of Tong et al. [12], Lo et al. [26], Wang X et al. [23], Wang W et al. [14] and meta-analysis by Czumbel et al. [27] also showed NPS a

**Table 3.  *ORF1b* positivity of various samples for a maximum number of days in daily monitoring of 15 follow up cases.**

| Patient No. | ORF1b positivity for maximum number of days during treatment | | | |
|:---:|:---:|:---:|:---:|:---:|
| | Combined swab | NPS | OPS | Sputum |
| P1 | 5 | 4 | 4 | 4 |
| P2 | 4 | 3 | 2 | 3 |
| P3 | 2 | 2 | 1 | 1 |
| P4 | 4 | 3 | 3 | 3 |
| P5 | 4 | 4 | 4 | 4 |
| P6 | 4 | 4 | 3 | 3 |
| P7 | 7 | 5 | 5 | 5 |
| P8 | 3 | 3 | 3 | 3 |
| P9 | 4 | 2 | 2 | 2 |
| P10 | 9 | 9 | 8 | 9 |
| P11 | 2 | 1 | 1 | 1 |
| P12 | 5 | 4 | 4 | 4 |
| P13 | 6 | 5 | 5 | 5 |
| P14 | 5 | 4 | 4 | 4 |
| P15 | 4 | 3 | 3 | 3 |
| Average days positivity | 4.5 | 3.7 | 3.4 | 3.6 |

better specimen for detecting SARS-CoV-2. The higher positivity rate of NPS could be correlated to higher viral load in nasopharynx than other anatomical sites/specimens.

Our study did not detect SARS-CoV-2 in clinical specimens of serum, plasma and urine. Earlier reported study too not found nCoV in either blood or urine specimens [28]. Chan et al. [24], Lo et al. [26], Wang W et al. [14] and reported meta-analysis showed negative results in urine specimen [17]. In contrast, only Tong et al. [12] had detected nCoV in urine, albeit with a low positivity rate of 16.3%. Low positivity rate of 12.5%, 1% and 0.9% was also reported in blood specimen by Tong et al. [12], Bwire et al. [17] and Wang W et al. [14], respectively. Chan et al. [24] found only one positive among three tested serum specimens while there was no positivity detected in plasma specimen. The number of specimens tested by Chan et al. [24] is too low to draw any relevant conclusion. Thus, it is advocated to conduct more studies on larger cohort to evaluate the role of blood and its components in diagnosing SARS-CoV-2 by rRT-PCR and its potential role in transmitting the virus. Ours and earlier published analysis for the absence of SARS-CoV-2 in urine showed that it is not shed from the urogenital system. Among the optimal specimen, earlier published meta-analysis found BAL with higher positive rate of detection (91.8%) of SARS-CoV-2 followed by rectal swabs (87.8%), sputum (68.1%), nasopharyngeal swab (45.5%), feces (32.8%), oropharyngeal swab (7.6%), and blood samples (1.0%) [17]. Another meta-analysis on respiratory samples found sputum with a significantly higher positive rate of detection of nCoV followed by NPS and OPS [18]. Tong et al. [12], on the other hand, found NPS having highest positive detection rate of nCoV among other specimen spectrum of BAL, NPS, OPS, sputum, urine, blood, stool, anal swab and corneal secretion (2.99%) [12]. Rao et al. [11], found saliva a better specimen than paired NPS+ OPS swab. Thus, it is inferred, that an ideal appropriate specimen varied in above-discussed studies. However, considering, the fact that more studies find NPS an ideal specimen in the identification of nCoV, our suggested combined swab would be the most appropriate specimen in the pandemic situation due to fulfilling the parameters of applicability in the variable clinical spectrum of the disease, easy accessibility in a larger group of patients, lesser risk hazard to health worker and higher detection rate than NPS.

**Table 4. Comparative evaluation of our finding with earlier studies.**

| Study | Nature | No. of Samples | BAL | Sputum | NPS | OPS | Both swab | Feces | Blood | Urine | Rectal/Anal swab | Serum | Plasma | FBB | ET | Nasal swab | Random saliva |
|---|---|---|---|---|---|---|---|---|---|---|---|---|---|---|---|---|---|
| Wang W et al. [14] | Cross sectional | Tested | 15 | 104 | 8 | 398 | - | 153 | 307 | 72 | - | - | - | 13 | - | - | - |
| | | Positive | 14 | 75 | 5 | 126 | - | 44 | 3 | 0 | - | - | - | 6 | - | - | - |
| Wang X et al. [23] | Cross sectional | Tested | - | - | 353 | 353 | 353 | - | - | - | - | - | - | - | - | - | - |
| | | Positive | - | - | 67 | 27 | 76 | - | - | - | - | - | - | - | - | - | - |
| Xu et al. [34] | Prospective | Tested | - | - | 49 | - | - | - | - | - | 49 | - | - | - | - | - | - |
| | | Positive | - | - | 22 | - | - | - | - | - | 43 | - | - | - | - | - | - |
| Lo et al. [26] | Prospective | Tested | - | 1 | 84 | - | - | 79 | - | 49 | - | - | - | - | - | - | - |
| | | Positive | - | 1 | 57 | - | - | 46 | - | 0 | - | - | - | - | - | - | - |
| Chan et al. [24] | Case series | Tested | - | 3 | 5 | 3 | - | 4 | - | 5 | - | 3 | 4 | - | - | - | - |
| | | Positive | - | 2 | 4 | 2 | - | 0 | - | 0 | - | 1 | 0 | - | - | - | - |
| Chen et al. [33] | Retrospective | Tested | - | 206 | 167 | - | - | 64 | - | - | - | - | - | - | - | - | - |
| | | Positive | - | 155 | 65 | - | - | 17 | - | - | - | - | - | - | - | - | - |
| Liu R et al. [25] | Cross sectional | Tested | 5 | 57 | 4818 | - | - | - | - | - | - | - | - | - | - | - | - |
| | | Positive | 4 | 28 | 1843 | - | - | - | - | - | - | - | - | - | - | - | - |
| Xie et al. [28] | Cross sectional | Tested | - | - | - | 19 | - | 19 | 19 | 19 | - | - | - | - | - | - | - |
| | | Positive | - | - | - | 9 | - | 8 | 0 | 0 | - | - | - | - | - | - | - |
| Liu M et al. [10] | Cross sectional | Tested | - | - | 47 | 47 | - | - | - | - | 47 | - | - | - | - | 47 | - |
| | | Positive | - | - | 26 | 9 | - | - | - | - | 1 | - | - | - | - | 23 | - |
| Rao et al. [11] | Cross sectional | Tested | - | - | - | - | 562 | - | - | - | - | - | - | - | - | - | 562 |
| | | Positive | - | - | - | - | 48 | - | - | - | - | - | - | - | - | - | 60 |
| Tong et al. [12] | Cross sectional | Tested | 15 | 382 | 463 | 39 | - | 262 | 40 | 135 | 98 | - | - | - | - | - | - |
| | | Positive | 7 | 61 | 297 | 10 | - | 32 | 3 | 12 | 8 | - | - | - | - | - | - |
| Zhang H et al. [13] | Cross sectional | Tested | - | 97 | 97 | 97 | - | - | - | - | - | - | - | - | 14 | - | - |
| | | Positive# | - | 79.2 | 37.5 | 20.8 | - | - | - | - | - | - | - | - | 13 | - | - |
| Our study | Cross sectional | Tested | - | 130 | 130 | 130 | 130 | - | - | - | - | - | - | - | 14 | - | - |
| | | Positive | - | 82 | 119 | 94 | 130 | - | - | - | - | - | - | - | 13 | - | - |

*This study did not show number of specimens detected. # Figures represent percentage.

The present study also showed a high positive rate of COVID-19 in males than females, as infected males were almost twice that of females. The various earlier studies and meta-analysis too observed a higher male susceptibility than females to COVID-19 [14, 23, 29]. The prominent possible factors included higher expression of angiotensin-converting enzyme -2 (ACE-2) attachment receptors in males than females, higher incidence of heart disease, high blood pressure in males, immunological differences driven by hormones and X chromosome and behavioral difference of increased personal habits of smoking and consuming alcohol etc. Higher susceptibility of males was further precipitated by the reported epidemiological observation that males have a more casual approach than females in appropriate compliance to wearing face mask, performing hand hygiene and maintaing social distancing practices [30, 31].

In terms of correlating lower Ct value with high viral load, our study showed detection of high viral load in the combined swab than other specimens. The individual NPS had the lowest Ct values in comparison to other individual specimens. This finding has also been corroborated by Wang W et al. [14], and Zou et al. [32], who also found higher viral load in NPS than OPS.

Our study also exclusively assessed the most appropriate clinical specimen to monitor the COVID-19 patients' treatment during their follow-up. Combined swabs exhibited longer duration of detection of nCoV as it is the last specimen during treatment follow-up to become negative among all seven types of specimens tested. This finding indicates that the combined swabs were the most appropriate specimen to assess virus clearance among the follow-up patients and thus equip the clinician in patient management and discharge. Data search found one brief report on 22 patients showing that sputum and feces remain positive even after NPS turned negative [33]. Another study on ten pediatric COVID 19 patients by Xu et al. [34] showed that rRT-PCR of rectal swabs was persistently positive after their NPS had become negative.

Novelty of the present study lies in the finding of combined swabs as an ideal specimen in both diagnosis and monitoring of treatment follow-up of symptomatic patients to better assess virus clearance, which eventually helps in discharge of truly recovered patients. This finding has clinical implication as early negative results with other specimens in follow-up investigation can give pseudoimpression of virus clearance leading to the potential risk of transmission of the COVID-19 infection in case if such patients are discharged. Among the published literature, Rao et al. [11], although found lower sensitivity of paired NPS + OPS swab versus saliva in asymptomatic patient, the difference of study group leaves a scope of further study involving both symptomatic and asymptomatic patients. Nevertheless, the probable reason for higher positivity using combined swab in our study than Rao et al. [11] could be the more viral load in symptomatic than in asymptomatic patients and strict adherence to sample collection in morning without nasal and throat wash.

Although stool and rectal/anal swab specimen were not tested in our study, few studies showing detection of nCoV in these specimens indicate them as a potential specimen for diagnosis [5, 10, 12, 14, 17, 23]. These findings suggest that nCoV resist the human gut acidic medium and could be transmitted through the fecal route. Presence of nCoV in stool is also substantiated by evidential presence of its receptors ACE 2 in enterocytes. Nevertheless, the correlation of this potential biological specimen for diagnosis and probability of the virus transmission through feco-oral route deserves further evaluation, since the virus viability in stool has not yet fully explored except Wang W et al. [14] reporting live nCoV from the stool specimen.

The limitation of present study is non-evaluation of some of the other potential specimens like BAL, FBB, saliva, stool and rectal swab. Obtaining BAL and FBB was avoided since their collection requires an invasive procedure that may pose high-risk aerosol exposure to health care workers. The feces and rectal/anal swab are also not primarily indicated considering the respiratory droplet being the commonest established transmission mode of nCoV. Clinical specimen of feces and rectal/anal swab cannot be considered an optimal specimen considering the limitation of difficulty in collection, transport and processing in comparison to respiratory specimens. Another specimen of saliva has a variable reported finding. Apart from Rao et al. [11] who found saliva a better specimen, earlier reported meta-analysis and review had found saliva to be of low sensitivity than NPS [27, 35]. Saliva has also not been recommended by either WHO or our regional authorities (ICMR) in their interim guidance for detection of SARS-CoV-2 [19, 36]. Therefore, saliva was not included in our study. We also could not correlate Ct values of *ORF1b* and *RdRP* with clinical features or disease course because most of the patients' detailed clinical information was not available.

Thus, this study concludes that NPS and OPS alone may miss some SARS-CoV-2 positive cases and hence should not be used exclusively as the sole specimen for diagnosis. Clinical specimen of serum, plasma and urine also should not be used for detection of SARS-CoV-2 by rRT-PCR. This study strongly recommends combined swab as the preferred clinical specimen

for detection of SARS-CoV-2 to establish diagnosis of COVID-19. The combined swab may also be considered the most appropriate specimen for monitoring of the follow-up cases to provide a better prognostic indicator of viral clearance during treatment. Therefore, the combined swab specimen has tremendous translational value for defining the recommendation in testing guidelines. Implementing the same globally will help manage and control the pandemic, as it is the need of the hour. Lower Ct in combined and NPS specimen also indicates towards the indirect evidence of posterior nasopharynx as the primary nCoV colonization site. Since blood, serum, plasma and urine were negative for the presence of nCoV in our study, the other transmission routes were not confirmed in the study and requires more studies with larger sample size for specific conclusive finding.

## Author Contributions

**Conceptualization:** Sanjay Singh Negi, Anudita Bhargava, Padma Das, Nitin M. Nagarkar.

**Data curation:** Deepa Gandhi, Priyanka Singh, Sanjay Singh Negi, Anudita Bhargava, Nitin M. Nagarkar.

**Formal analysis:** Kuldeep Sharma, Pragya Aggarwala, Deepa Gandhi, Sanjay Singh Negi.

**Funding acquisition:** Priyanka Singh.

**Investigation:** Anuniti Mathias, Priyanka Singh, Somya Sharma, Ajoy Behra.

**Methodology:** Kuldeep Sharma, Pragya Aggarwala, Deepa Gandhi, Anuniti Mathias, Priyanka Singh, Somya Sharma, Sanjay Singh Negi.

**Project administration:** Anuniti Mathias, Ujjwala Gaikwad, Archana Wankhede.

**Resources:** Anuniti Mathias, Archana Wankhede, Ajoy Behra.

**Software:** Deepa Gandhi.

**Supervision:** Sanjay Singh Negi, Anudita Bhargava, Archana Wankhede.

**Validation:** Priyanka Singh, Ujjwala Gaikwad.

**Visualization:** Somya Sharma, Ujjwala Gaikwad, Archana Wankhede.

**Writing – original draft:** Sanjay Singh Negi.

**Writing – review & editing:** Sanjay Singh Negi, Anudita Bhargava, Padma Das, Ujjwala Gaikwad, Nitin M. Nagarkar.

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
