## [Decision Letter · Decision Letter 0]

27 Jan 2021

PONE-D-20-38439

Comparative analysis of various clinical specimens in detection of SARS-CoV-2 using rRT-PCR in new and follow up cases of COVID-19 infection: quest for the best choice

PLOS ONE

Dear Dr. Negi,

Thank you for submitting your manuscript to PLOS ONE. After careful consideration, we feel that it has merit but does not fully meet PLOS ONE’s publication criteria as it currently stands. Therefore, we invite you to submit a revised version of the manuscript that addresses the points raised during the review process.

I have received the reviews of your manuscript. While your paper addresses an interesting question, the reviewer stated several concerns about the presentation as well as the readability of the manuscript and did not recommend publication in its present form. Please see reviewer’s insightful comments below and these comments need to be addressed carefully. I agreed with the concerns raised by the reviewer and have some comments that need to be addressed (see specific comments) In addition, the quality of the language needs to be improved. There are awkward sentences, typo, etc. throughout the manuscript making it hard to assess the scientific content.  Please have a fluent, preferably native, English-language speaker thoroughly copyedit your manuscript for language usage, spelling, and grammar.  If you do not know anyone who can do this, we suggest you use a professional language editing or copyediting service.  

Specific comments:

The format of the manuscript needs to be consistent, e.g. change Table 1 to (Table 1), Fig.1 to (Fig. 1), etc.Abstract, ling 8:  Please spell out Please spell out NPS (nasopharyngeal swabs) and OPS (oropharyngeal swabs) here since this is the first time NPS and OPS were mentioned. What is combined swab? Do the authors mean combined nose throat swab? Please clarify.Line 33: “…in extreme of age.”, Did the authors mean younger and older or just older? This needs rephrasing and clarify.Line 69:  Change “Sputum, Plasma, Serum, Urine and Tracheal aspirate” to “sputum, plasma, serum, urine and tracheal aspirate”.Line 130, change 2x buffer to 12.5 μl 2x buffer.

Line 97:  Need approval #.

Line 137 – 139, 2.4 Gold standard:  This is not how the gold standard is defined.  I suggest deleting this section.Line 266:  The positive rate should be 0.9% not 0.009%.

We look forward to receiving your revised manuscript.

Kind regards,

Baochuan Lin, Ph.D.

Academic Editor

PLOS ONE

Journal Requirements:

3. Thank you for including the following ethics statement on the submission details page:

'Study is approved form Institutional Ethical committee.

Name of committe: IEC-AIIMS, Riapur.

Approval number: AIIMSRPR/IEC/2020/536'

Please also include  the specific name of your ethics committee and the approval number in the Methods section of your manuscript."

7. Please include your tables as part of your main manuscript and remove the individual files. Please note that supplementary tables should be uploaded as separate "supporting information" files.

Reviewers' comments:

Reviewer's Responses to Questions

**Comments to the Author**

1. Is the manuscript technically sound, and do the data support the conclusions?

Reviewer #1: Partly

2. Has the statistical analysis been performed appropriately and rigorously? 

Reviewer #1: Yes

3. Have the authors made all data underlying the findings in their manuscript fully available?

Reviewer #1: Yes

4. Is the manuscript presented in an intelligible fashion and written in standard English?

Reviewer #1: No

5. Review Comments to the Author

Reviewer #1: Appropriate sample collection and detection by rRT-PCR is very important for the diagnosis of novel coronavirus (nCoV) disease (COVID-19). The authors compared the diagnostic value of samples from different sources of the body.

NPS, OPS, combined swab, sputum, plasma, serum and urine were investigates in 924 samples before initiation of any treatment in 130 COVID-19 cases. They also collected 665 follow up samples from NPS, OPS, combined swab, sputum, plasma, serum and urine from 15 randomly selected cases. Virus detection was performed by rRT-PCR.

Combined swab showed positivity rate of 100 % followed by NPS (91.5%), OPS (72.3%), sputum (63%) while nCoV was found undetected in urine, plasma and serum specimens.

The analysis of 665 follow-up specimens showed that the combined NPS and OPS results gave best diagnostic value.

They concluded that combined swab detections should be applied for detecting and monitoring f COVID-19 by rRT-PCR. They suggested that their lower Ct in combined and NPS swab indicated the primary nCoV colonization site is the posterior nasopharynx.

The paper investigates an extremely interesting and important topic. But unfortunately, its serious shortcomings should be handled before publication.

Comments:

1. It is clear when one looks at the cited works of the manuscript that the reference search of the authors ended in mid June 2020. No later paper is cited except a WHO guideline. This is unacceptable in this exploding area of research.

Both that introduction and the discussion must include recent comparative meta-analyses on the subject. In this respect, the Discussion should also emphasize the novelty of the present work compared to those.

Bwire GM, Majigo MV, Njiro BJ, Mawazo A. Detection profile of SARS-CoV-2 using RT-PCR in different types of clinical specimens: A systematic review and meta-analysis. J Med Virol. 2021 Feb;93(2):719-725. doi: 10.1002/jmv.26349. Epub 2020 Aug 2. PMID: 32706393; PMCID: PMC7404904.

Mohammadi A, Esmaeilzadeh E, Li Y, Bosch RJ, Li JZ. SARS-CoV-2 detection in different respiratory sites: A systematic review and meta-analysis. EBioMedicine. 2020 Sep;59:102903. doi: 10.1016/j.ebiom.2020.102903. Epub 2020 Jul 24. PMID: 32718896; PMCID: PMC7380223.

2. Key original papers having similar aims as the present work should also be quoted and compared to the present data in the Discussion:

Liu M, Li Q, Zhou J, Ai W, Zheng X, Zeng J, Liu Y, Xiang X, Guo R, Li X, Wu X, Xu H, Jiang L, Zhang H, Chen J, Tian L, Luo J, Luo C. Value of swab types and collection time on SARS-COV-2 detection using RT-PCR assay. J Virol Methods. 2020 Dec;286:113974. doi: 10.1016/j.jviromet.2020.113974. Epub 2020 Sep 16. PMID: 32949663; PMCID: PMC7493793.

Tong Y, Bao A, Chen H, Huang J, Lv Z, Feng L, Cheng Y, Wang Y, Bai L, Rao W, Zheng H, Wu Z, Qiao B, Zhao Z, Wang H, Li Y. Necessity for detection of SARS-CoV-2 RNA in multiple types of specimens for the discharge of the patients with COVID-19. J Transl Med. 2020 Nov 2;18(1):411. doi: 10.1186/s12967-020-02580-w. PMID: 33138834; PMCID: PMC7605325.

Zhang H, Chen M, Zhang Y, Wen J, Wang Y, Wang L, Guo J, Liu C, Li D, Wang Y, Bai J, Gao G, Wang S, Yang D, Yu F, Yan L, Wan G, Zhang F. The Yield and Consistency of the Detection of SARS-CoV-2 in Multiple Respiratory Specimens. Open Forum Infect Dis. 2020 Aug 26;7(10):ofaa379. doi: 10.1093/ofid/ofaa379. PMID: 33072810; PMCID: PMC7499703.

Rao M, Rashid FA, Sabri FSAH, Jamil NN, Seradja V, Abdullah NA, Ahmad H, Aren SL, Ali SAS, Ghazali M, Manaf AA, Talib H, Hashim R, Zain R, Thayan R, Amran F, Aris T, Ahmad N. COVID-19 screening test by using random oropharyngeal saliva. J Med Virol. 2021 Jan 4. doi: 10.1002/jmv.26773. Epub ahead of print. PMID: 33393672.

3. The authors suggested that their lower Ct in combined and NPS swab indicated the primary nCoV colonization site is the posterior nasopharynx. They did not provide any direct evidence for this. Therefor this should be removed from the major findings and conclusion. However, the indirect evidence that they provided should be discussed and compared to data of others received by other methodologies.

4. The data and the details of Figure 4 are simply invisible. Downloaded high resolution does not help on this a lot. For visibility and clarity, this figure should be completely redrawn.

5. The limitations of the study should be discussed in the Discussion in more details. For example, saliva is one of the most promising diagnostic sample. This should be discussed. At least the following meta-analysis should be cited and briefly credited in the discussion:

Czumbel LM, Kiss S, Farkas N, Mandel I, Hegyi A, Nagy Á, Lohinai Z, Szakács Z, Hegyi P, Steward MC, Varga G. Saliva as a Candidate for COVID-19 Diagnostic Testing: A Meta-Analysis. Front Med (Lausanne). 2020 Aug 4;7:465. doi: 10.3389/fmed.2020.00465. PMID: 32903849; PMCID: PMC7438940.

6. The English language of the paper needs extensive revision by a professional language editor. Particularly, many sentences are very long, complicated, therefore, hard to understand.

6. PLOS authors have the option to publish the peer review history of their article (what does this mean?). If published, this will include your full peer review and any attached files.

Reviewer #1: **Yes: **Gabor Varga

---

## [Author Response · Author response to Decision Letter 0]

8 Feb 2021

We sincerely thank PLOS editor , editorial team and reviewer for their efforts in criticially reviewing our resereach article to provide various valuable suggestion. Incorporating the same will definitely help us in improving our article.

---

## [Decision Letter · Decision Letter 1]

5 Mar 2021

PONE-D-20-38439R1

Comparative analysis of various clinical specimens in detection of SARS-CoV-2 using rRT-PCR in new and follow up cases of COVID-19 infection: quest for the best choice

PLOS ONE

Dear Dr. Negi,

Thank you for submitting your manuscript to PLOS ONE. After careful consideration, we feel that it has merit but does not fully meet PLOS ONE’s publication criteria as it currently stands. Therefore, we invite you to submit a revised version of the manuscript that addresses the points raised during the review process.

While the manuscript is scientifically sound, there are format, typos and awkward sentences through out the manuscript that needs to be corrected (I have attached a file containing examples of my suggestion). We suggest you thoroughly copyedit your manuscript for language usage, spelling, and grammar. If you do not know anyone who can help you do this, you may wish to consider employing a professional scientific editing service.

Whilst you may use any professional scientific editing service of your choice, PLOS has partnered with both American Journal Experts (AJE) and Editage to provide discounted services to PLOS authors. Both organizations have experience helping authors meet PLOS guidelines and can provide language editing, translation, manuscript formatting, and figure formatting to ensure your manuscript meets our submission guidelines. To take advantage of our partnership with AJE, visit the AJE website (http://learn.aje.com/plos/) for a 15% discount off AJE services. To take advantage of our partnership with Editage, visit the Editage website (www.editage.com) and enter referral code PLOSEDIT for a 15% discount off Editage services. If the PLOS editorial team finds any language issues in text that either AJE or Editage has edited, the service provider will re-edit the text for free."

We look forward to receiving your revised manuscript.

Kind regards,

Baochuan Lin, Ph.D.

Academic Editor

PLOS ONE

Journal Requirements:

Reviewers' comments:

Reviewer's Responses to Questions

**Comments to the Author**

1. If the authors have adequately addressed your comments raised in a previous round of review and you feel that this manuscript is now acceptable for publication, you may indicate that here to bypass the “Comments to the Author” section, enter your conflict of interest statement in the “Confidential to Editor” section, and submit your "Accept" recommendation.

Reviewer #1: All comments have been addressed

2. Is the manuscript technically sound, and do the data support the conclusions?

Reviewer #1: Yes

3. Has the statistical analysis been performed appropriately and rigorously? 

Reviewer #1: Yes

4. Have the authors made all data underlying the findings in their manuscript fully available?

Reviewer #1: Yes

5. Is the manuscript presented in an intelligible fashion and written in standard English?

Reviewer #1: Yes

6. Review Comments to the Author

Reviewer #1: (No Response)

7. PLOS authors have the option to publish the peer review history of their article (what does this mean?). If published, this will include your full peer review and any attached files.

Reviewer #1: **Yes: **Gabor Varga

---

## [Author Response · Author response to Decision Letter 1]

9 Mar 2021

We have tried to adequately address all the suggestions and comments made by reviewer and journal editorial office.

---

## [Editor Report · Decision Letter 2]

11 Mar 2021

PONE-D-20-38439R2

Comparative analysis of various clinical specimens in detection of SARS-CoV-2 using rRT-PCR in new and follow up cases of COVID-19 infection: quest for the best choice

PLOS ONE

Dear Dr. Negi,

Thank you for submitting your manuscript to PLOS ONE. After careful consideration, we feel that it has merit but does not fully meet PLOS ONE’s publication criteria as it currently stands. Therefore, we invite you to submit a revised version of the manuscript that addresses the points raised during the review process.

The revised version is scientifically sound except a few format, typos, grammatic and reference errors that need to be corrected (please see attached PDF file).

We look forward to receiving your revised manuscript.

Kind regards,

Baochuan Lin, Ph.D.

Academic Editor

PLOS ONE
---

## [Author Response · Author response to Decision Letter 2]

16 Mar 2021

We have done all required changes as suggested by the journal.

---

## [Editor Report · Decision Letter 3]

18 Mar 2021

Comparative analysis of various clinical specimens in detection of SARS-CoV-2 using rRT-PCR in new and follow up cases of COVID-19 infection: quest for the best choice

PONE-D-20-38439R3

Dear Dr. Negi,

We’re pleased to inform you that your manuscript has been judged scientifically suitable for publication and will be formally accepted for publication once it meets all outstanding technical requirements.

Kind regards,

Baochuan Lin, Ph.D.

Academic Editor

PLOS ONE
---

## [Editor Report · Acceptance letter]

26 Mar 2021

PONE-D-20-38439R3 

Comparative analysis of various clinical specimens in detection of SARS-CoV-2 using rRT-PCR in new and follow up cases of COVID-19 infection: quest for the best choice 

Dear Dr. Negi:

I'm pleased to inform you that your manuscript has been deemed suitable for publication in PLOS ONE. Congratulations! Your manuscript is now with our production department. 

Kind regards, 

on behalf of

Dr. Baochuan Lin 

Academic Editor

PLOS ONE